# Machine Learning Approach for Automated Detection of Irregular Walking Surfaces for Walkability Assessment with Wearable Sensor

**DOI:** 10.3390/s23010193

**Published:** 2022-12-24

**Authors:** Hui R. Ng, Isidore Sossa, Yunwoo Nam, Jong-Hoon Youn

**Affiliations:** 1Department of Computer Science, University of Nebraska Omaha, Omaha, NE 68182, USA; 2Community and Regional Planning, University of Nebraska-Lincoln, Lincoln, NE 68588, USA

**Keywords:** wearable accelerometer, machine learning, gait analysis, walkability, sidewalk surface assessment

## Abstract

The walkability of a neighborhood impacts public health and leads to economic and environmental benefits. The condition of sidewalks is a significant indicator of a walkable neighborhood as it supports and encourages pedestrian travel and physical activity. However, common sidewalk assessment practices are subjective, inefficient, and ineffective. Current alternate methods for objective and automated assessment of sidewalk surfaces do not consider pedestrians’ physiological responses. We developed a novel classification framework for the detection of irregular walking surfaces that uses a machine learning approach to analyze gait parameters extracted from a single wearable accelerometer. We also identified the most suitable location for sensor placement. Experiments were conducted on 12 subjects walking on good and irregular walking surfaces with sensors attached at three different locations: right ankle, lower back, and back of the head. The most suitable location for sensor placement was at the ankle. Among the five classifiers trained with gait features from the ankle sensor, Support Vector Machine (SVM) was found to be the most effective model since it was the most robust to subject differences. The model’s performance was improved with post-processing. This demonstrates that the SVM model trained with accelerometer-based gait features can be used as an objective tool for the assessment of sidewalk walking surface conditions.

## 1. Introduction

Walking is one of the most popular and cheapest forms of physical activity. A high level of walking in a community leads to various public health, environmental, and economic benefits [1,2,3,4]. Therefore, maintaining walkable neighborhoods has garnered increased interest. Walkability measures are an indication of the extent of the friendliness of a neighborhood-built environment to pedestrian walking, and walkability assessment is used to evaluate such friendliness. It also considers pedestrians’ feelings [2]. The availability and condition of sidewalks are significant predictors of perceived safety and general satisfaction in the pedestrian environment [5]. Therefore, assessing sidewalks is an integral component of walkability assessment tools [6].

To take pedestrians’ perceptions into account, one of the common sidewalk assessment practices is through pedestrian interviews or surveys [7]. However, these responses can be biased and lack expert insights. Another common practice for sidewalk assessment is on-site inspection performed by experts to identify violations of pre-defined regulations [8]. Due to the ineffectiveness and inefficiency of these practices, alternative advanced methods have been proposed to automate sidewalk assessment using infrastructure-based data such as street-view images, videos, or GIS technologies. However, these methods do not capture subjects’ interactions with their external walking environments.

With the advancement of wearable sensors, it is possible to measure and analyze human physiological responses to surrounding environments. The portability and affordability of these sensors makes it possible to develop a real-time sidewalk condition monitoring system that incorporates the personal characteristics of pedestrians. A few studies have been conducted on using machine learning techniques for sidewalk assessment utilizing smartphone acceleration data collected from subjects. To our knowledge, no studies have been conducted on utilizing machine learning techniques that incorporate gait analysis to assess sidewalk walking surface conditions with a single accelerometer. Furthermore, no study has determined the most suitable placement of a single wearable sensor to assess walking surface conditions.

In this study, we propose a novel classification framework for the walkability domain that uses machine learning to analyze gait features extracted from a single wearable accelerometer for the detection of irregular walking surfaces. We also identify the most suitable location on the human body to place a single accelerometer for irregular walking surface detection. We determine the most effective machine learning model along with an optimal subset of gait features that best discriminates between good and irregular walking surfaces, which can be implemented as a real-world application for identification of problematic walking areas in a continuous manner. Furthermore, the optimal subset of gait features helps to understand gait parameters most affected by irregular walking surfaces.

The remainder of the paper is structured as follows: First, we present a review of related works. Next, the experimental design and the research methodology utilized in this work are discussed. The results are then provided followed by a discussion on the findings and implication of this study.

## 2. Related Works

### 2.1. Traditional Approaches for Sidewalk Assessment

A pedestrian survey is one of the common approaches used to detect sidewalk defects [7]. This method is subjective, costly, and ineffective in providing a detailed analysis of these defects [7]. Field inspections by trained inspectors are another common method practiced by governmental agencies for identification of regulatory violations [8]. Researchers have also proposed methods to improve current sidewalk assessment practices. Sousa et al. proposed a method to evaluate sidewalks using field measurements [9]. Another study also proposed computing the Pavement Condition Index based on surveys designed for the purpose of sidewalk condition assessment [10]. However, these methods are labor-intensive and unreliable. Furthermore, they are not scalable to larger cities.

### 2.2. Advanced Approaches for Sidewalk Assessment

Many advanced methods have been proposed to automatically assess roadways based on urban data such as images, videos, or GIS technologies [11,12,13,14,15,16]. Several studies have proposed automated machine learning approaches to estimate roadway anomalies using smartphone sensors mounted on vehicles [17,18,19,20]. One study [21] measured the surface roughness of pedestrian and bicycle lanes with a smartphone mounted on bicycles. However, these data sources were unable to answer the question of how the human body responds to roadway or sidewalk surfaces [22]. Individuals respond differently to the same surroundings based on their characteristics; therefore, their variability must be analyzed [7].

Only a few methods have been proposed to assess sidewalk or roadway surfaces in an automated fashion that incorporates human bodily responses using machine learning techniques. Miyata et al. [23] estimated barriers and obstacles using a machine learning model trained with acceleration data from pedestrians’ smartphones. Another study proposed an approach to detect road bumps with acceleration data from smartphones that are placed on cyclists’ pants [24]. Kobayashi et al. estimated sidewalk surface types using acceleration data from smartphones placed on pedestrians [25]. They achieved that by training a Random Forest model. Kobayashi et al. expanded their study to train a convolutional neural network to estimate sidewalk surface types [26]. In Kobayashi et al.’s studies [25,26], the models trained with window-based features extracted from smartphone data were not robust to subject differences.

Several studies have discovered that it is feasible to utilize pedestrians’ physiological responses measured with wearable sensors placed at various locations to detect different sidewalk characteristics or defects [7,27,28]. There are also gait analysis studies conducted to examine the effects of irregular and uneven surfaces on human gait [29,30,31,32,33]. Therefore, we assume that accelerometer-based gait features can reduce noise and overfitting compared to commonly used window-based features from smartphones, which can lead to more generalizable models towards subject differences.

### 2.3. Sensor Location

Several studies have placed sensors at different locations to examine the effects of the external environment and irregular surfaces on subjects’ gait parameters. In examining the correlation between pedestrians’ gait patterns and the built environment, Kim et al. utilized acceleration data from the right ankle to compute gait parameters [27]. Another study evaluated the effects of even and irregular surfaces on basic gait parameters and head and hip accelerations [33]. The researchers expanded on that study to the older population [34].

Employing sensors at multiple locations for real-time continuous monitoring is not practical. Furthermore, adding data from multiple locations in the analysis causes the process to be computationally intensive [35]. In addition, multi-sensor approaches have potential user acceptance and user friendliness issues. Therefore, we evaluate sensor placement at the head, the hip, and the right ankle for our study to identify the most suitable location for distinguishing good and irregular walking surfaces.

## 3. Materials and Methods

The workflow of incorporating gait analysis with machine learning techniques is illustrated in Figure 1. It consists of four steps: data pre-processing, feature extraction, feature selection, and modelling.

### 3.1. Data Collection

Twelve healthy subjects—eight males and four females—were recruited to participate in the study. The subjects wore three accelerometers at different locations on the body. As shown in Figure 2, the head accelerometer was mounted on the back of a cap while the hip accelerometer was affixed at the back of the hip. The right ankle accelerometer was mounted on the outer side of the subject’s shoe.

The linear accelerations of the body were measured using Mbient sensors (MetaMotionR, Mbient Lab, San Francisco, CA, USA). These tri-axial accelerometers were configured to sample data at 100 Hz. The experiment was set up to mimic a real-world sidewalk setting. A stretch of well-paved, smooth, and levelled walking paths at The Peter Kiewit Institute of the University of Nebraska at Omaha, denoted with starting and ending points, was selected for the experiment. Within that stretch, four irregular walking surface segments were created: a grass-covered surface segment, a surface segment with obstructions, an uneven surface segment, and a debris-covered surface segment. These walking segments are typical walking surfaces that are likely to be encountered in the real-world. Figure 3 illustrates the good and irregular walking segments. Subjects were instructed to walk on the path at their normal walking speed from the starting point to the ending point and back to the starting point on the same path. Figure 4 visualizes the walking pattern of a subject with a duration of five seconds (500 data points) in the form of raw acceleration readings captured by sensors placed at the three different locations.

### 3.2. Data Pre-Processing

The *X*, *Y*, and *Z* axes of the ankle accelerometer represent signals in Vertical (V), Anteroposterior (AP), and Mediolateral (ML) directions, respectively, as illustrated in Figure 5. As for accelerometers located on the hip and head, the *X*, *Y*, and *Z* axes represent V, ML, and AP directions, respectively.

For data pre-processing, the timestamp data from the three accelerometers were first synchronized using the synchronize function in MATLAB R2022a (Mathworks, Natick, MA, USA). Then, the raw acceleration data were labelled by referring to the video recordings. Since the aim was to analyze gait patterns to detect irregular walking surfaces, all irregular walking surface scenarios were grouped into one Irregular class. Before extracting gait features, stride segmentation was first performed on the raw data using the AP directional acceleration of the right ankle accelerometer for accurate stride recognition [36]. The stride distribution for each walking surface label is displayed in Table 1.

### 3.3. Feature Extraction

We extracted features that have demonstrated to be useful in mobility studies [36,37]. First, 20 base features were extracted, of which the descriptions and formulas are presented in Table 2. The VM5, VMD, LVMD, VVMD, AVMD, VM30, LVM30, VVM30, and AVM30 features captured the characteristics of the initial 5% of the stance, double stance, and mid-stance phases of a gait cycle. We also captured the directional impulse and the magnitude of the initial loading phase through features LHM, LHSD, VHM, VHSD, AHM, and AHSD. The characteristics of a whole stride were also extracted to understand the stride magnitude, directional magnitude of the stride, and stride duration. For each of those 20 base features, we also computed the variability of those features between strides. This summed to a total of 40 gait features for each sensor.

### 3.4. Most Suitable Sensor Location Selection

We first determined the most suitable sensor location for irregular walking surface detection. To ensure that the most suitable sensor location was subject agnostic, we used a protocol that iterates through all 12 subjects to leave one subject out as the test set and the remaining subjects as the training set. For each iteration, the three sensor locations were evaluated using a stratified five-fold cross-validation (CV) method repeated five times to compare the classification performance of each of the classification models mentioned in Section 3.6 across all sensor locations. The full set of features extracted for all three locations was fed into the models. This resulted in a set of 12 metrics per location for each classifier. A one-way ANOVA test was performed, followed by a Tukey post-hoc multiple comparison test, to assess whether the location of the accelerometer had a statistically significant impact on the classification performance of each model.

### 3.5. Feature Selection

Since incorporating additional features is computationally costly for sensor-based systems and affects model interpretability, there is a need to systematically identify the optimal subset of the original extracted feature set of the most suitable location. We compared two popular feature selection techniques in this study: Elastic Net (ENET) and Maximum Relevance and Minimum Redundancy (MRMR). Prior to selecting features, the extracted features were normalized to zero mean and scaled to unit variance.

#### 3.5.1. Elastic Net (ENET)

Multicollinearity occurs when there is a strong correlation between features that could lead to excessively complex models and overfitting. ENET [38] is a popular regularization and feature selection method that deals with multicollinearity between features. It combines the advantages of two well-known penalties known as Least Absolute Shrinkage and Selection Operator (LASSO) and Ridge. The ENET regularization approach solves the problem through the following equation:(1)$$min_{\beta }\left[\frac{1}{n}\overset{n}{\underset{i=1}{\sum }}{\left(y_{i}-x_{i}^{T}\beta \right)}^{2}+\lambda \left(\left(1-\alpha \right)\frac{{\left\|\beta \right\|}_{2}^{2}}{2}+\;\alpha {\left\|\beta \right\|}_{1}\right)\;\right]$$
where $y_{i}$ and $x_{i}^{T}=\left(x_{i1},\ldots ,x_{ip}\right)$ are the response and predictors of the *i*-th stride, respectively. ${\left\|\beta \right\|}_{1}$ and ${\left\|\beta \right\|}_{2}$ are the LASSO and ridge penalty, respectively. α is the mixing parameter that determines the ratio of LASSO and ridge penalty with the range of [0, 1], with α = 0 being ridge penalty, while α = 1 being LASSO. λ is the regularization parameter, and the choice of both λ and α is crucial to selecting important variables.

The optimal α and λ parameters were selected using a stratified five-fold CV with a grid search. This was repeated 12 times using the leave-one-subject-out as the test set protocol. Gait studies on mobility have demonstrated that good prediction metrics can be achieved with only eight to 10 features while balancing model complexity and computational cost [35,37]. Hence, we selected only the top 10 features at each iteration. Not all the top 10 features selected were the same at each iteration. Therefore, we compiled and profiled the top 10 features selected for all iterations by the frequency of each feature being selected, which quantified the importance of the features.

To identify the optimal number of ENET top features to include in the final ENET feature subset, based on the feature profile, we added the features one-by-one starting with the most important feature, and ran our models using a repeated stratified five-fold CV to obtain the performance metric. The classification performance across all models would improve until they plateau as more important features were added. We determined the optimal number of features threshold based on the model that yielded the best metrics across all iterations of feature adding, since we are evaluating ENET and MRMR based on the best metric that each method can achieve. The threshold would be the point where the rate of improvement of the metric decreases. We identified that threshold objectively using an algorithm called Kneedle [39]. This is a knee detection algorithm applicable to a wide range of cost-benefit analysis problems based on a formal mathematical definition and was implemented using a Python package called Kneed.

#### 3.5.2. Maximum Relevance and Minimum Redundancy (MRMR)

MRMR [40,41] is a well-known method that selects features by considering the relevance of the features for predicting the target variable and the redundancy among the selected features. We chose the MRMR F-test correlation quotient MRMR-FCQ for our study among the variants because it has demonstrated better performance in terms of computation time and robust accuracy [41].

In the MRMR process, we need to specify the number of top *k* features we would like to select. The MRMR framework iterates *k* times until the top *k* features are selected. At each iteration, each feature would be scored based on this formula:(2)$$f^{FCQ}\left(X_{i}\right)=F\left(Y,X_{i}\right)/\left[\frac{1}{\left|S\right|}\underset{X_{s}\in S}{\sum }\rho \left(X_{s},X_{i}\right)\right]$$
where $X_{i}\left(i\in \;1,2,\ldots ,40\right)$ denotes one of the total 40 features, *Y* is the class label, *S* is the set of selected features, *|S|* denotes the number of features, $X_{s}$ ∈ *S* is one of the features in the feature set, and $X_{i}$ is one of the features not in the selected set. The relevance between a given feature $X_{i}$ and *Y* is measured with an F-statistic as denoted by *F*(*Y*, $X_{i}$). As for redundancy between selected features $X_{s}$ and features yet to be selected $X_{i}$, it is measured with Pearson correlation ρ($X_{s}$, $X_{i}$). The feature with the highest score was added to *S*.

We set *k* = 10 at each leave-one-subject-out iteration to select the top 10 features and compiled all the top features selected across all iterations, similar to the ENET procedure. Then, we identified the optimal number of MRMR top features to include in the final MRMR feature subset using the same approach described for ENET. The results are presented in. Finally, we compared the two feature subsets of MRMR and ENET. The optimal feature subset would be the one that produced the best metric with the smaller number of features.

### 3.6. Classification Models

We compared five classifiers in this study: Support Vector Machine (SVM), Random Forest (RF), Logistic Regression (LR), Ada Boost (ADA), and Extreme Gradient Boosting (XGB). SVM, RF, LR, and ADA are classification algorithms commonly used for gait analysis. XGB [42] was selected because it is a widely recognized algorithm that has consistently won many machines learning challenges. These classifiers are implemented using Python’s scikit-learn module [43]. A brief description for each of the models is as follows:SVM: SVM maps features onto a high-dimensional feature space using kernels for non-linear class boundaries and finds a hyperplane that distinctively classifies the samples well [44]. We configured our SVM to use a radial basis function kernel and searched for the optimal cost parameter, the kernel coefficient value, and gamma that gave us the best metric.RF: RF is an ensemble learning algorithm that builds a number of decision trees to solve classification tasks. Each tree is built with a subset of the training data, with each node in the tree split using the best randomly selected sample of features. A majority vote from the decision trees is taken to obtain the final classification [44]. For this model, we optimized the number of trees in the forest, the maximum depth of each tree, the minimum number of samples required in each node split, the number of features to consider while searching for the best split, and the minimum samples required for a leaf node.LR: LR is a widely used classification model for binary classification. It estimates the association of one or more features with a binary response variable utilizing the maximum likelihood of finding the regression coefficients for each feature to minimize the distance between the predicted probability of each class to the actual class [44].ADA: ADA is a boosting ensemble learning algorithm that takes a weighted combination of multiple weak learners into a strong classifier [45]. These weak learners are one-level decision trees called decision stumps.XGB: XGB is a gradient boosting ensemble learning classifier. It utilizes the gradient descent algorithm on decision trees for sequentially building stronger learners from weak learners by minimizing a loss function [42]. We optimized the maximum depth of each tree, the learning rate at each iteration during training, the number of trees, the fraction of samples to train for each tree, the fraction of features to be sampled randomly per tree, and the fraction of features to be used in each node for each tree.

To select the most effective model, we trained the top three classifiers observed during feature selection using the best feature subset. We used the leave-one-subject-out as the test set protocol to evaluate the subject-wise generalizability of our final models. As each iteration would yield a different training set, we tuned the hyperparameters for classifiers at each iteration using a randomized search with a stratified five-fold CV. The optimized models were then evaluated using the test set. The classifiers were compared using the test metrics averaged across all subjects.

### 3.7. Performance Metric

Since the class imbalance problem may bias certain performance metrics such as accuracy except for the area under the ROC curve [46], the Area Under the Curve (AUC) metric was used to evaluate the classifiers. The AUC metric is a single-value metric that summarizes the ROC curve to evaluate the performance of a classifier. The samples that are labelled as irregular walking surfaces, which is the minority class, are the positive class for our experiment.

### 3.8. Class Imbalance

Based on the label distribution in Table 1, our dataset was moderately imbalanced, as the proportion of irregular walking surfaces was only 35%. Since an imbalance in training data would cause classifiers to have poor performance on the minority class [46], we used re-sampling techniques to address that.

We used an over-sampling technique called Synthetic Minority Oversampling TEchnique (SMOTE) [47]. SMOTE creates synthetic examples of the minority class using the k-nearest neighbor algorithm to add samples to the minority class to balance the classes [47]. We configured SMOTE from Python’s imblearn package [48] to oversample the minority class with synthetic examples utilizing 5-nearest neighbor such that the class distribution for the majority and minority classes had a class ratio of 50:50.

## 4. Results

### 4.1. Most Suitable Sensor Location for Irregular Walking Surface Detection

Figure 6 shows the distribution of the resulting AUC from the leave-one-subject-out as the test set protocol. The AUC distribution for the right ankle location can be observed to be consistently better than the head and the hip locations.

To determine whether the impact of the location on the classification performance is statistically significant for each classifier, a one-way ANOVA test was performed. Table 3 displays the results of the ANOVA tests for each classifier. Based on the p-values, we can conclude that the location of the accelerometer has a significant impact on classification performance for all five classifiers. We then used the Tukey Pairwise comparison test to perform multiple comparisons.

Based on the results of Tukey’s pairwise test in Table 4 and the box plot, it can be concluded that the right ankle location was significantly different from the other two locations and produced the best performance for all classifiers. The head and hip locations had no significant impact on classification performance for all classifiers except for SVM. Therefore, we conclude that the ankle is the most suitable location to place an accelerometer to detect irregular walking surfaces.

### 4.2. Optimal Feature Subset

Since the right ankle accelerometer is the most suitable location for discriminating between good and irregular walking surfaces, we performed feature selection on the right ankle feature set for optimal feature subset selection.

#### 4.2.1. ENET Results

From the leave-one-subject-out as the test set protocol, the optimal λ ranged from 0.001 to 0.01 while α ranged from 0.0 to 0.9. Figure 7 displays the frequency profile of the top 10 selected features compiled from the leave-one-subject-out protocol.

It resulted in 17 features, since the top 10 features selected in each iteration differ. Some features were selected as frequently as other features, which means they have the same importance and usefulness in discriminating between good and irregular walking surfaces. Therefore, to determine the optimal number of top ENET features to include in the final ENET feature subset, we added the features batch-by-batch instead of one-by-one, starting from the most important feature, and ran them through all five models. As observed in Figure 8, when more features are added, the AUC would improve until a point when adding more features only contributed to slight improvements in classification performance or the AUC would decrease.

We then used the Kneedle algorithm to objectively identify the threshold that balances the trade-off between model performance and training computational cost. Since different curves produced by different classifiers would yield different thresholds, we used the Kneedle algorithm to identify the optimal number of ENET top features based on the model that yields the best metric across all iterations, which is RF. Based on Figure 9, the top eight features selected by ENET were able to achieve an AUC of 87%. The final ENET feature subset consisted of its top eight features.

#### 4.2.2. MRMR Results

The frequency profile of the top 10 selected features by MRMR compiled from the leave-one-subject-out as the test set protocol is shown in Figure 10.

There are 17 features in total. To determine the optimal number of top MRMR features to include in the final MRMR feature subset, we added one feature batch at a time moving down the importance rank and evaluated them with our models. The results are illustrated in Figure 11. It can be observed that SVM outperforms RF when there are 13 or more features.

Therefore, we ran the Kneedle algorithm on the performance results of both models. As we can see from Figure 12 and Figure 13, the selected thresholds were the same for both models at 13 features. The top 13 features of MRMR produced the best performance using SVM, which is at an AUC of 85%. Hence, the final MRMR feature subset consisted of the top 13 features as shown in Figure 10.

#### 4.2.3. ENET and MRMR Comparison

The best classification performance achieved by the full feature set and feature subsets selected by ENET and MRMR is shown in Table 5. The best AUC attained by ENET’s eight features is similar to the AUC obtained when using all 40 features. Furthermore, the ENET feature subset was able to achieve better classification performance with fewer number of features compared to MRMR’s performance.

Looking at Table 6, when classifiers were trained with ENET’s top 8 features, they achieved better classification results than MRMR’s top eight features in general. The same can also be observed when comparing the top 13 features of ENET with MRMR. Since we are optimizing by selecting the feature subset that can yield the best classification performance with the least number of features, the optimal feature subset chosen was the top eight features selected by ENET. Additionally, taking a closer look at Figure 7 and Figure 10, VM was selected as one of the most important features for both methods, which indicates that VM is a strong predictor. Comparing the classifiers in Table 6, we can also identify SVM, RF, and XGB as the top three classifiers for the purpose of discriminating between good and irregular walking surfaces.

### 4.3. Most Effective Model

We trained our top three classifiers identified during feature selection with the optimal feature subset. The test results for all three classifiers for each subject iteration including the average test results are shown in Table 7. SVM achieved the highest average test AUC. It was also the most robust to subject differences compared to RF and XGB since the test AUC of SVM was always greater than 70% when tested with gait patterns of a subject unseen during training and had the lowest standard deviation. Hence, we can conclude that SVM is the most effective model for irregular walking surface detection.

### 4.4. Post-Processing

We observed that when subjects were walking on irregular walking surfaces, interrupted gait occurred intermittently among normal gaits. Not every stride was interrupted. Up until this point, we detected irregular walking surfaces on a per stride basis by predicting if each stride occurred on an irregular surface or not. Therefore, we assumed that if we combined several consecutive strides’ walking surface predictions by a classifier and took the average of the predicted probability of occurring on an irregular walking surface of the combined gaits to produce a final prediction of the walking surface covered by the combined gaits, it would improve the classification metric. We use a sliding window approach to segment the per stride prediction obtained from classifiers for each subject. *k* consecutive strides were sampled into one segment to produce the final prediction for that segment, sliding to the next segment by the step size of one stride. The results are shown in Figure 14. As we increased *k*, the classification performance improved across all three classifiers, which confirmed our assumption. With post-processing, the average AUC of the most effective model, SVM, improved from 80% to 85%.

## 5. Discussion

In this study, we identified the most suitable placement of a single accelerometer for the purpose of discriminating between good and irregular walking surfaces, which is at the ankle. Then, we selected the optimal subset of gait features extracted from the raw accelerations of the ankle sensor to train several machine learning classifiers for comparison. The most effective model could detect irregular walking surfaces with satisfactory classification performance. Our results demonstrate the feasibility of utilizing wearable accelerometers and a machine-learning approach to differentiate good and irregular walking surfaces.

Our results show that the accelerometer placed at the right ankle had better capability of estimating walking surface conditions compared to accelerometers placed at the hip and the head. It improved the classification performance of various models. This could be because changes in the magnitude of accelerations were smaller at the head and the hip compared to the ankle when subjects were walking on irregular walking surfaces. Therefore, changes in underlying stride parameters could be better captured when the sensor was placed at the ankle. Furthermore, as the ankle sensor was closer to the ground, a subtle gait adaptation for each stride when walking on irregular walking surfaces can be captured. Therefore, we recommend that a wearable acceleration sensor be placed at the ankle when implementing the proposed method in a real-time walking surface condition assessment application.

This study also determined an optimized feature set that balances computational cost for a real-time sensor-based system, model interpretability, and classification performance. The optimized feature set consisted of eight features: VM, VVMD, VMD, VM30, AVM, LVMD, VHSD, and VVM. The feature VM was found to be the strongest predictor. AVM and VVM, which characterize the whole stride acceleration of the ankle sensor in different directions, were also selected. They may have been selected because subjects would adapt their whole gait to maintain their stability while walking on irregular walking surfaces. Three out of eight features selected characterize the double-stance phase: VVMD, LVMD, and VMD. Since the double-stance phase occurs when the body weight transitions from one limb to another [37], stability is affected during this phase, which could explain the importance of these features.

SVM was found to be the most generalizable subject-wise despite the limited number of subjects. The results have demonstrated that SVM was more robust to individual differences compared to RF and XGB. Its classification performance was also further improved with post-processing by combining the predicted probability of several strides to produce final predictions. This means that when the proposed method is implemented as an application, at least five seconds of data need to be measured since the average stride time is around 1.10 s based on our data.

Since current sidewalk condition assessment usually relies on trained experts from governmental agencies or voluntary participation of residents or pedestrians, the intervals between assessments are long due to staffing and budget limitations [7]. On the other hand, automated assessment methods utilizing urban data do not reflect pedestrians’ responses to sidewalk surface conditions while walking. In contrast, the proposed machine learning approach can be implemented as a tool to assess sidewalk walking surface conditions based on pedestrians’ gait patterns measured with a single accelerometer. This approach is highly practical since it can detect irregular walking surfaces with a high classification accuracy using only a single wearable sensor. As a pedestrian is walking with the wearable sensor, the sidewalk walking surface conditions can be continuously assessed by analyzing his or her real-time bodily response data. It will reduce the time and cost required for on-site inspection while eliminating human biases during the assessment. Unlike automated approaches using urban data, the proposed approach addresses the user-oriented, effectiveness, and efficiency aspects of sidewalk surface condition assessment.

One of the limitations of this study is that the experiment was not conducted in a real-world walkable and less walkable neighborhood. Subjects could exhibit different walking patterns in an experimental setting. The other limitation is that the sample size in terms of the number of subjects is small. To address that, we drew conclusions repeatedly by iteratively leaving one subject out as the test set to evaluate our methods. Despite these limitations, this study is the first step towards an objective and automated assessment of sidewalk walking surface conditions of a neighborhood.

For future work, we will develop a deep learning model for the same purpose with a single accelerometer placed on the ankle. We are considering using a Long Short-Term Memory network, which is widely used in the domain of human activity recognition. Future experiments will be conducted in a real-world walkable and less walkable neighborhood with more subjects. The best machine learning classifier and the optimal feature set can be used to develop an irregular walking surface detection model utilizing real-world data. Additionally, GPS data can also be incorporated to locate and identify the cause of problematic walking surface areas.

## 6. Conclusions

In this study, we proposed a novel method for irregular walking surface detection using machine learning techniques and accelerometer-based gait features extracted from a single sensor. We also identified the most suitable location for sensor placement among the three locations commonly used to measure the effect of external environment on subjects’ gait parameters, the optimal feature set, and the most effective classifier for irregular walking surface detection. Our results indicate that the most suitable location for sensor placement is a subject’s ankle. We also find that SVM is the most effective model because it is the most generalizable and can achieve satisfactory classification performance with limited data. Furthermore, we also demonstrate that classification performance can be improved by taking several consecutive strides into account to make predictions. In conclusion, our results support the potential application of the proposed method as an objective tool for assessing the sidewalk walking surface condition of a neighborhood.

## Figures and Tables

**Figure 1 sensors-23-00193-f001:**
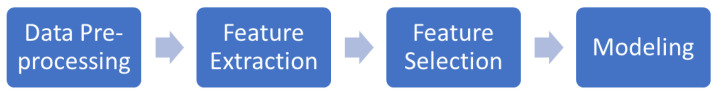
Workflow of gait analysis with machine learning.

**Figure 2 sensors-23-00193-f002:**
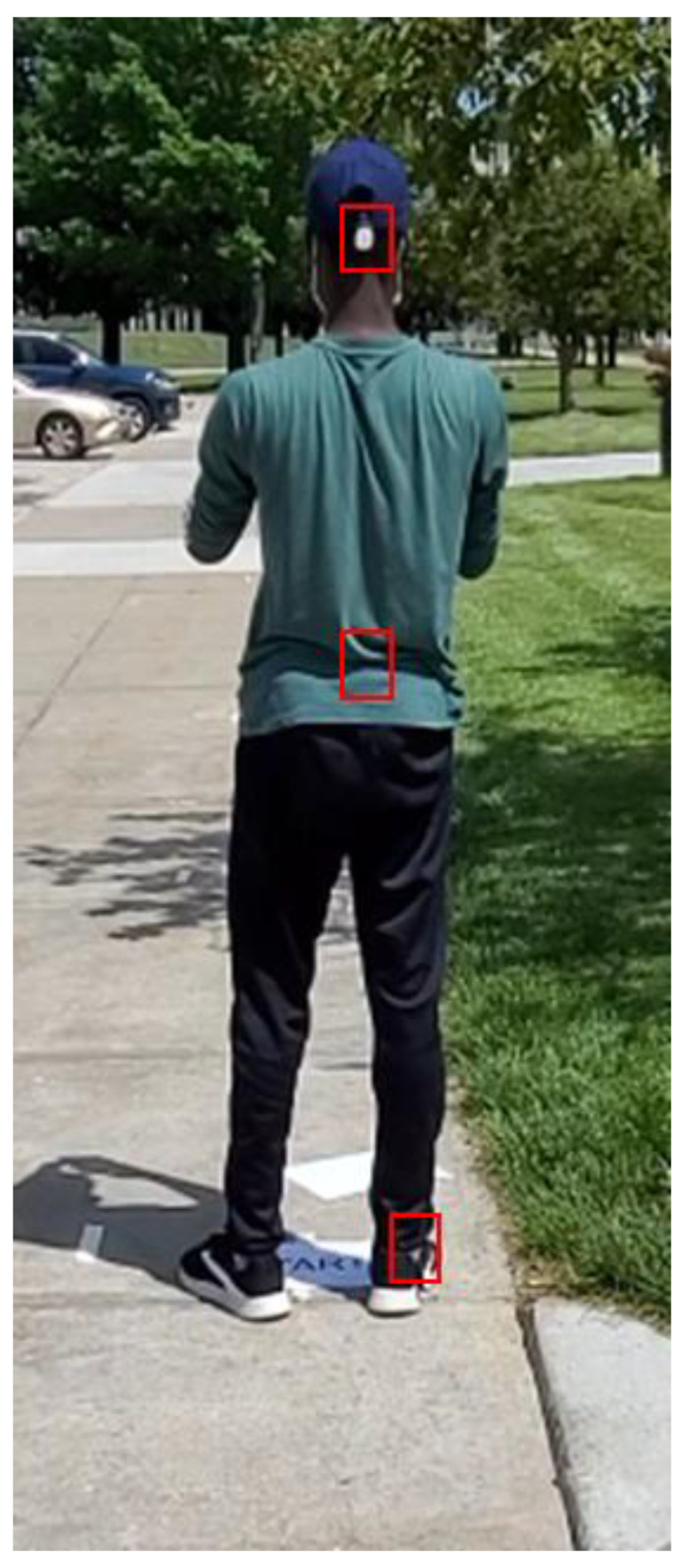
Placement of sensors at three locations on a subject.

**Figure 3 sensors-23-00193-f003:**
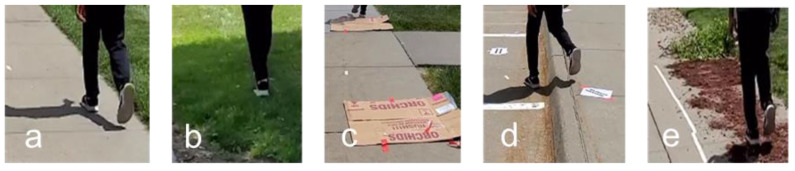
Good and typical irregular walking surfaces: (**a**) Good, well-paved; (**b**) Grass-covered (Irregular); (**c**) Obstructions (Irregular); (**d**) Uneven (Irregular); and (**e**) Debris-covered (Irregular).

**Figure 4 sensors-23-00193-f004:**
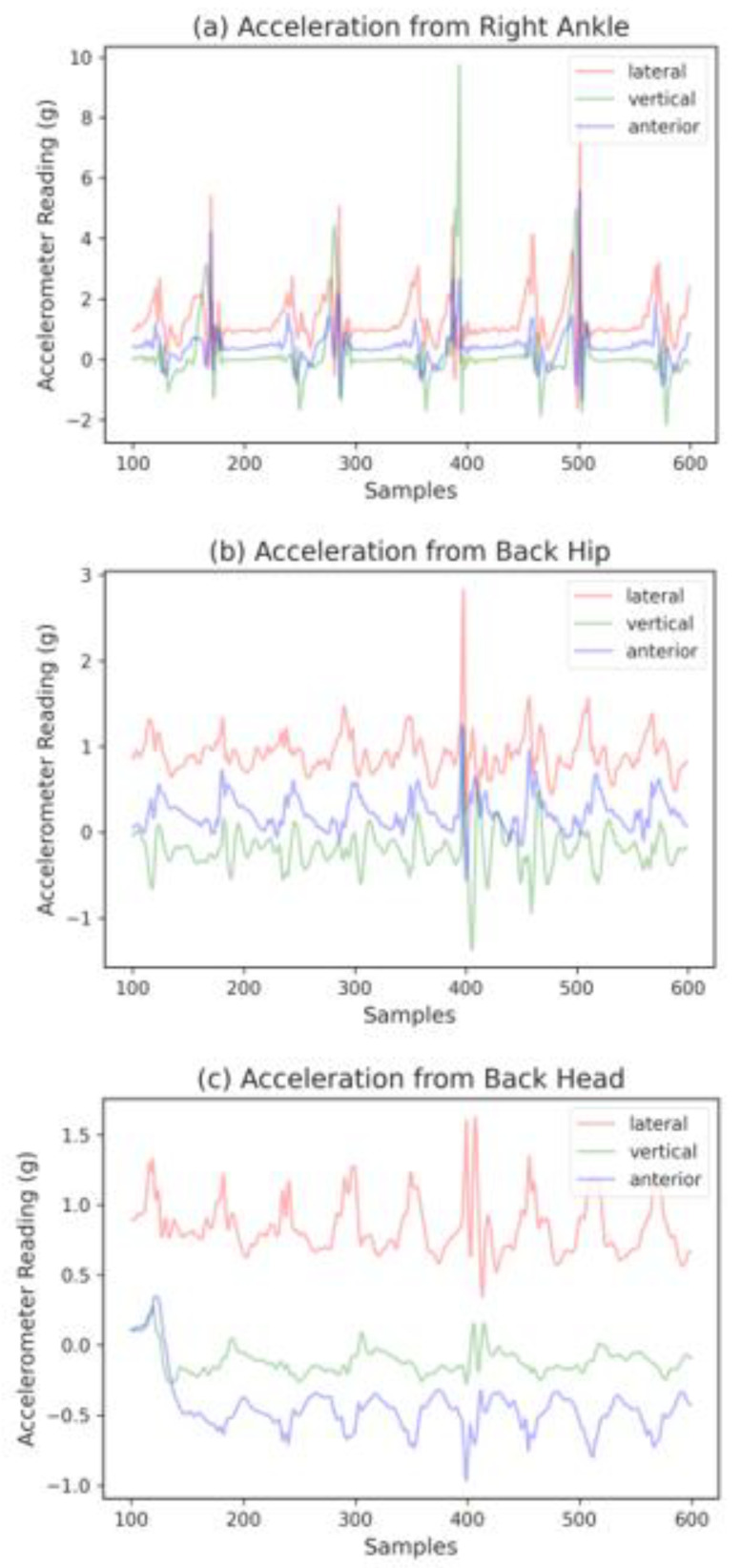
Five- second raw walking acceleration data of a subject measured by sensors placed at the right ankle, the back of the hip, and the back of the head.

**Figure 5 sensors-23-00193-f005:**
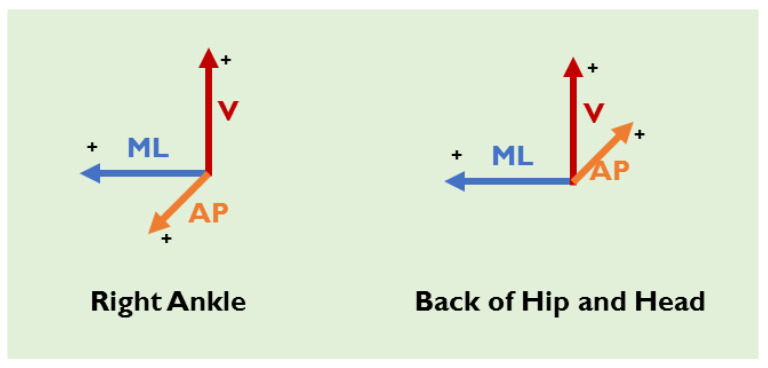
Accelerometers’ directions for all three locations.

**Figure 6 sensors-23-00193-f006:**
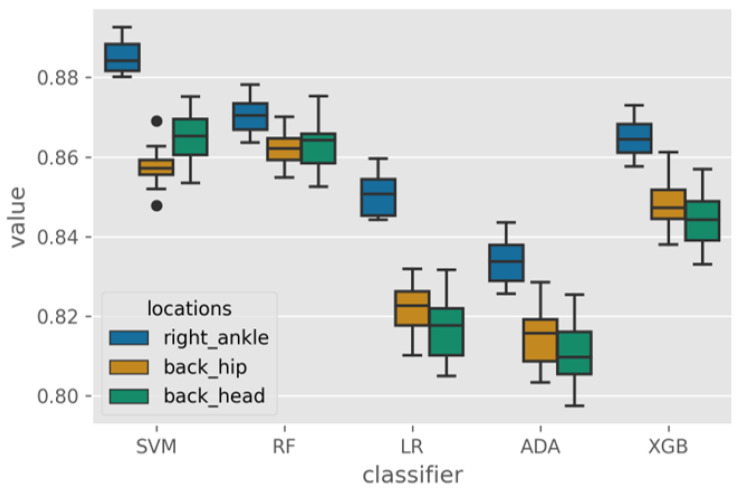
Box plot of the AUC across classifiers and locations.

**Figure 7 sensors-23-00193-f007:**
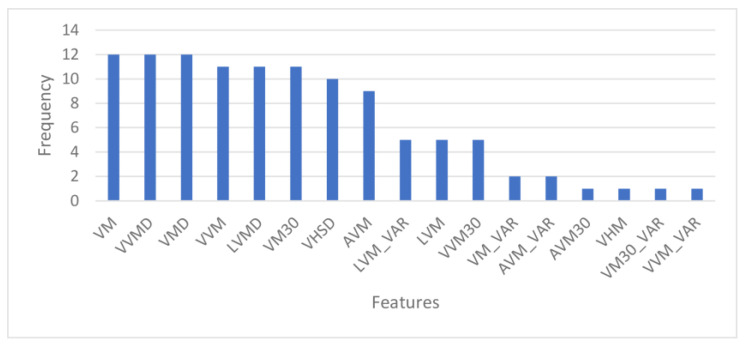
Frequency of features selected by ENET.

**Figure 8 sensors-23-00193-f008:**
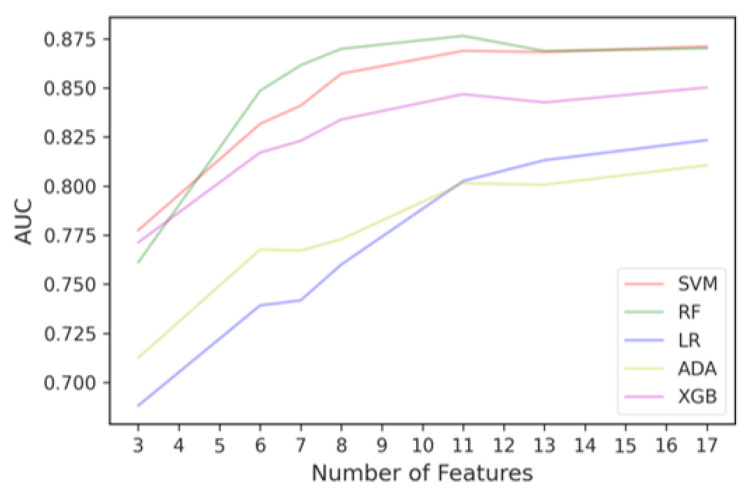
Classification performance curve from adding important features selected by ENET batch-by-batch.

**Figure 9 sensors-23-00193-f009:**
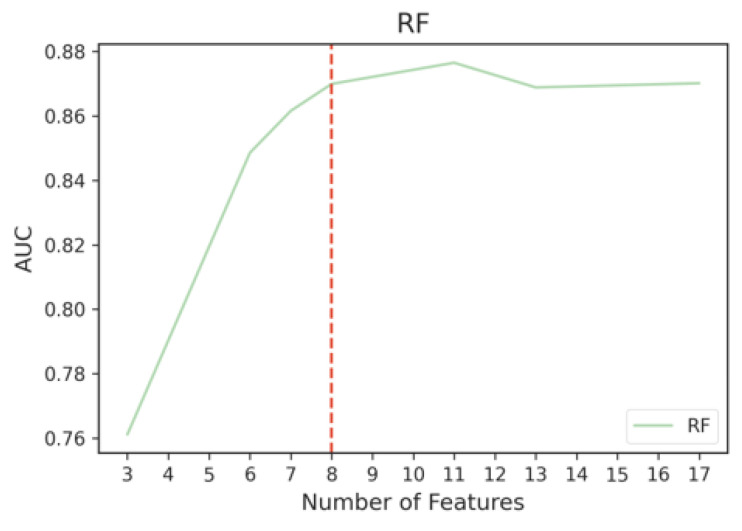
Cutoff point for RF where performance tapers off when adding more ENET features.

**Figure 10 sensors-23-00193-f010:**
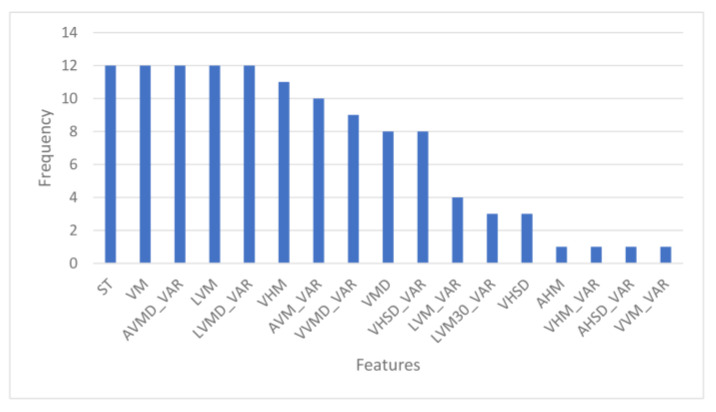
Frequency of features selected by MRMR.

**Figure 11 sensors-23-00193-f011:**
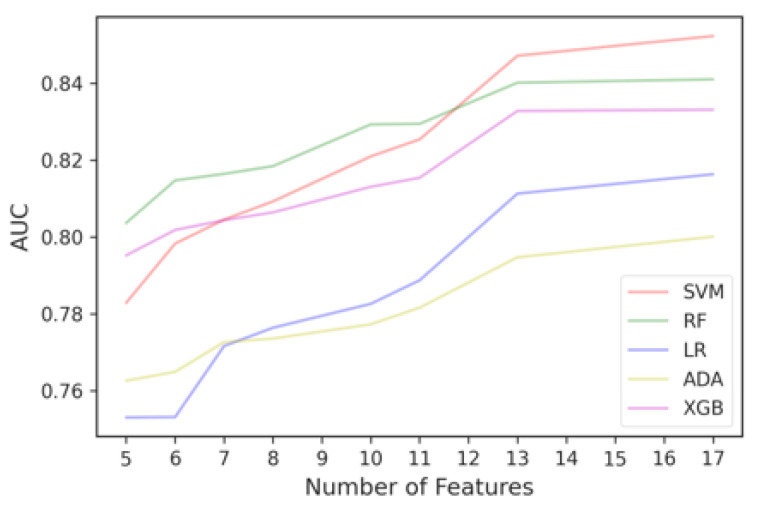
Classification performance curve from adding important features selected by MRMR batch-by-batch.

**Figure 12 sensors-23-00193-f012:**
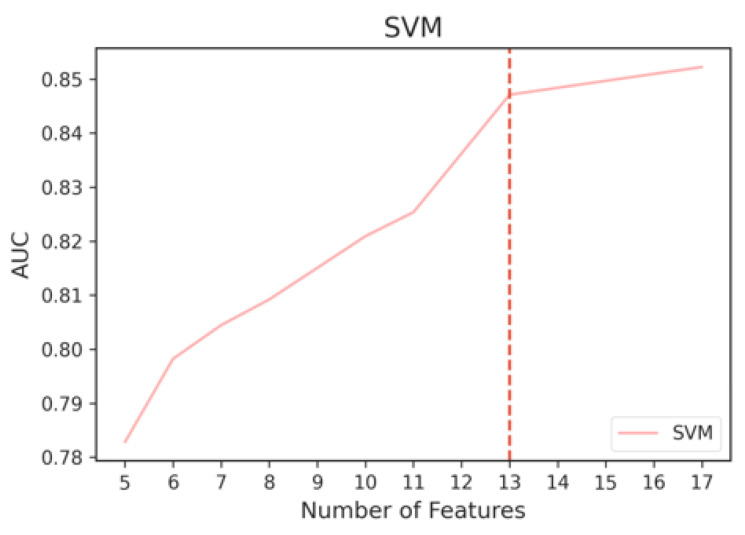
Cutoff point for SVM where performance tapers off when adding more MRMR features.

**Figure 13 sensors-23-00193-f013:**
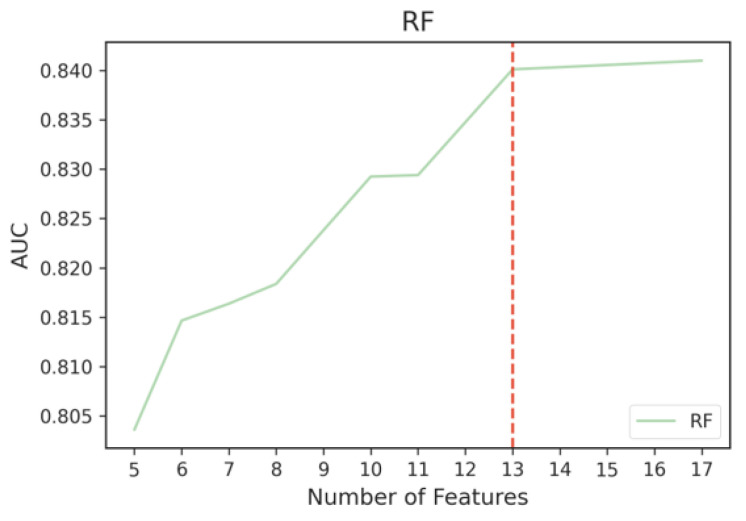
Cutoff point for RF, where performance tapers off when adding more MRMR features.

**Figure 14 sensors-23-00193-f014:**
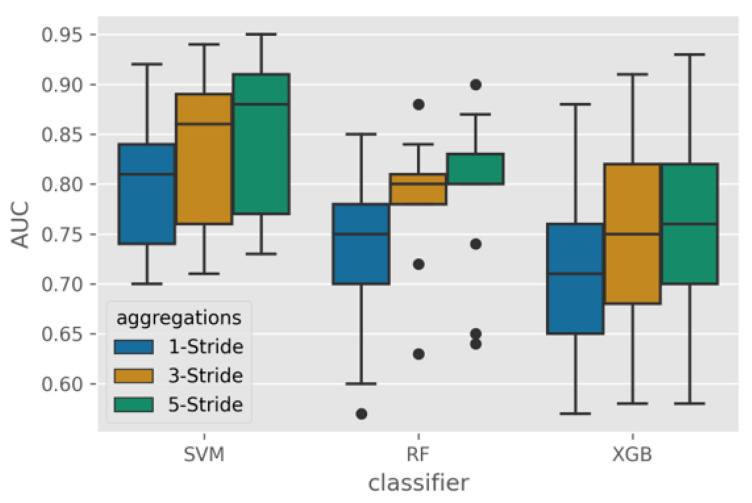
AUC distribution of test subjects using increasing stride aggregations for prediction for each classifier.

**Table 1 sensors-23-00193-t001:** Walking surface label distribution.

Label	Count	Percentage
Good	3774	65%
Irregular	1995	35%
Total	5769	100%

**Table 2 sensors-23-00193-t002:** Description of features extracted.

Feature	Description	Formula
VM	Vector magnitude for the whole stride	$\sqrt{ML^{2}+V^{2}+AP^{2}}$ for the whole stride vectors
VM5	Vector magnitude for the initial 5% of the stride	$\sqrt{ML^{2}+V^{2}+AP^{2}}$ for the initial 5% of the stride vectors
LVM	Vector magnitude of the ML direction for the whole stride	$\sqrt{ML^{2}}$ for the whole stride vectors
VVM	Vector magnitude of the V direction for the whole stride	$\sqrt{V^{2}}$ for the whole stride vectors
AVM	Vector magnitude of the AP direction for the whole stride	$\sqrt{AP^{2}}$ for the whole stride vectors
VMD	Vector magnitude for the double stance	$\sqrt{ML^{2}+V^{2}+AP^{2}}$ for ±10% vectors around the heel-strike event
LVMD	Vector magnitude for the ML direction during the double stance	$\sqrt{ML^{2}}$ for ±10% vectors around the heel-strike event
VVMD	Vector magnitude for the V direction during the double stance	$\sqrt{V^{2}}$ for ±10% vectors around the heel-strike event
AVMD	Vector magnitude for the AP direction during the double stance	$\sqrt{AP^{2}}$ for ±10% vectors around the heel-strike event
VM30	Vector magnitude for the mid-stance	$\sqrt{ML^{2}+V^{2}+AP^{2}}$ for vectors from 30% of the gait cycle
LVM30	Vector magnitude for the ML direction during the mid-stance	$\sqrt{ML^{2}}$ for vectors from 30% of the gait cycle
VVM30	Vector magnitude for the V direction during the mid-stance	$\sqrt{V^{2}}$ for vectors from 30% of the gait cycle
AVM30	Vector magnitude for the AP direction during the mid-stance	$\sqrt{AP^{2}}$ for vectors from 30% of the gait cycle
LHM	Heel-strike magnitude in the ML direction	Max (ML) at the heel strike
LHSD	Std. dev. of ML acceleration during the initial 10% stride	Std (ML) of the initial 10% of the stride vectors
VHM	Heel-strike magnitude in the V direction	Max (V) at the heel strike
VHSD	Std. dev. of V acceleration during the initial 10% stride	Std (V) of the initial 10% of the stride vectors
AHM	Heel-strike magnitude in the AP direction	Max (AP) at the heel strike
AHSD	Std. dev. of the AP acceleration during the initial 10% stride	Std (AP) of the initial 10% of the stride vectors
ST	Stride time	Time between two consecutive heel strikes

**Table 3 sensors-23-00193-t003:** One-way ANOVA results for each classifier.

Classifier	ANOVA *p*-Value
SVM	2.08 × 10^−13^
RF	4.42 × 10^−4^
LR	2.10 × 10^−13^
ADA	2.64 × 10^−9^
XGB	7.49 × 10^−9^

**Table 4 sensors-23-00193-t004:** Tukey’s pairwise comparison for location pairs across classifiers.

Classifiers	Group 1	Group 2	Mean Difference	Adjusted *p*-Values	Lower	Upper	Reject
SVM	Head	Hip	−0.0076	0.0053	−0.0131	−0.0021	TRUE
SVM	Head	Ankle	0.0201	0.001	0.0145	0.0256	TRUE
SVM	Hip	Ankle	0.0277	0.001	0.0221	0.0332	TRUE
RF	Head	Hip	−0.0011	0.8658	−0.0063	0.0042	FALSE
RF	Head	Ankle	0.0077	0.0031	0.0024	0.0129	TRUE
RF	Hip	Ankle	0.0087	0.001	0.0035	0.014	TRUE
LR	Head	Hip	0.005	0.2	-0.002	0.012	FALSE
LR	Head	Ankle	0.0335	0.001	0.0265	0.0405	TRUE
LR	Hip	Ankle	0.0285	0.001	0.0215	0.0355	TRUE
ADA	Head	Hip	0.0043	0.3008	−0.0027	0.0113	FALSE
ADA	Head	Ankle	0.0235	0.001	0.0165	0.0305	TRUE
ADA	Hip	Ankle	0.0192	0.001	0.0121	0.0262	TRUE
XGB	Head	Hip	0.004	0.2956	−0.0025	0.0105	FALSE
XGB	Head	Ankle	0.0207	0.001	0.0142	0.0272	TRUE
XGB	Hip	Ankle	0.0167	0.001	0.0102	0.0232	TRUE

**Table 5 sensors-23-00193-t005:** Comparison of the number of features selected and performance of ENET and MRMR.

	All Features	ENET	MRMR
Number of Features	40	8	13
Best AUC	0.88	0.87	0.85

**Table 6 sensors-23-00193-t006:** Classification results for top eight and top 13 features of ENET and MRMR for all classifiers.

	SVM	RF	LR	ADA	XGB
ENET Top 8 Features	0.85	0.87	0.76	0.77	0.83
MRMR Top 8 Features	0.81	0.82	0.78	0.77	0.81
ENET Top 13 Features	0.87	0.87	0.81	0.8	0.84
MRMR Top 13 Features	0.85	0.84	0.81	0.79	0.83

**Table 7 sensors-23-00193-t007:** Classification results for each test subject for the top three classifiers.

Test Set Subject	SVM	RF	XGB
A	0.74	0.78	0.62
B	0.81	0.60	0.76
C	0.83	0.72	0.85
D	0.70	0.57	0.57
E	0.85	0.76	0.72
F	0.70	0.75	0.65
G	0.84	0.70	0.73
H	0.87	0.78	0.63
I	0.82	0.78	0.76
J	0.72	0.70	0.68
K	0.75	0.76	0.68
L	0.92	0.85	0.88
Std. Deviation	0.07	0.08	0.09
Average	0.80	0.73	0.71

## Data Availability

Not applicable.

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
