# Peer review of "Machine Learning Approach for Automated Detection of Irregular Walking Surfaces for Walkability Assessment with Wearable Sensor"

_sensors, 2022, doi:10.3390/s23010193_

Round 1
Reviewer 1 Report
In this paper, the authors pressents a kind of objective assessing method of sidewalks using wearable sensors. By implementing experiments and extracting gait parameter signals from the accelemeter sensors amounted on 12 participators, who walk on four different conditions of sidewalks, the authors deal with the information using machining learning methods. They figure out that the most proper location for wearing the sensor is the right ankle. In comparison of hip and head, right ankle can show the best gait features with post processing, and the Support Vector Machine (SVM) is the most suitable model compared to other classifiers for training the gait features to assess the surface conditions of sidewalks.
Overall, this paper is well-organized and should be rather interesting as it demonstrates a new type of objective assessment manner by applying wearable sensors. I think this manuscript can be acceptable for publication in Sensors upon minor revision addessing the following questions.
1. For better comparison, the authors should also provide with data acquired from sensors on the other location, besides the right ankle data in Figure 4.
2. The figures in the paper should be more clear, e.g., Figure 8, 9, 11, 12,13. The lines are not very identical.
3. The text format should be corrected, e.g., from line 414 to line 419.
4. It is not well-explained by the diamond shape dots in Figure 6 and 14.
5. Some obvious errors can be seen in the References, e.g., Ref. 47 and 48.
Author Response
Thank you so much for taking the time to review the manuscript and provide constructive feedback. Based on your review comments, we added the following changes.
- The manuscript has been proof read by a native English speaking colleague to improve the readability of the manuscript.
- (Comment #1) In the revised manuscript, Figure 4 shows raw walking acceleration data measured by sensors placed at the right ankle, at the back of the hip, and at the back of the head.
- (Comment #2) The quality of the figures (Figure 6, 8, 9, 11, 12, and 13) was improved according to your suggestion.
- (Comment #3) The text format error was corrected.
- (Comment #4) The diamond shape dots in Figure 6 and 14 represent outliers. To eliminate the confusion, we replace the diamond shapes with smaller dots in Figure 6 & 14.
- (Comment #5) The errors in the reference were corrected.
Thanks again for all constructive review comments.
Reviewer 2 Report
This research aims to develop a novel classification framework for detecting irregular walking surfaces using machine learning. The framework uses gait parameters extracted from a single wearable accelerometer, allowing for an objective and automated assessment of sidewalk surfaces. The study found that the most suitable location for sensor placement is at the ankle, and among the five trained classifiers, the Support Vector Machine (SVM) model was the most effective. The use of this machine learning approach could provide a more efficient and effective way to assess sidewalk conditions and support the development of walkable neighborhoods.
I like the paper, I didnt find any flaws, I recommend it for publication
Author Response
Thank you so much for taking the time to review our manuscript. We are so grateful for your positive review comments.
Reviewer 3 Report
It is a very well written paper. The idea is presented clearly. The methodology of the paper is also clear. I have a just a few minor comments. The authors need to spell check their paper thoroughly. Moreover, the quality of the figures used in the paper is not of the highest standard. So, authors should improve the quality of the figures.
Author Response
Thank you so much for taking the time to review the manuscript and provide constructive feedback. Based on your review comments, we added the following changes:
- The manuscript has been proof read by a native English speaking colleague to improve the readability of the manuscript.
- The quality of the figures (Figure 4, 6, 8, 9, 11, 12, and 13) was improved according to your suggestion.